# The Impacts of Forest Therapy on the Physical and Mental Health of College Students: A Review

**Mei He** [1,2], **Yuan Hu** [2], **Ye Wen** [2], **Xin Wang** [1], **Yawei Wei** [1], **Gonghan Sheng** [1] and **Guangyu Wang** [1,*]

1 Department of Forest Resources Management, University of British Columbia, 2329 West Mall, Vancouver, BC V6T 1Z4, Canada; mei.he@ubc.ca (M.H.); wangx222@mail.ubc.ca (X.W.); yawei.wei@ubc.ca (Y.W.); jerrys92@mail.ubc.ca (G.S.)

2 Jiangxi Academy of Forestry, 1629 Fenglin Road, Nanchang 330032, China; xqx2021@gmail.com (Y.H.); wenye0791@foxmail.com (Y.W.)

* Correspondence: guangyu.wang@ubc.ca

**Abstract:** The aim of this review is to investigate the impacts of various forest therapy activities on the physical and mental health of college students. Additionally, it evaluates the research methodologies and existing issues in current studies, providing an important agenda for future research. Research was conducted based on the Preferred Reporting Items for Systematic Reviews and Meta-Analyses (PRISMA). The findings suggest significant effects of forest therapy activities on the physiology and psychology of college students, notably improving the cardiovascular system, enhancing the immune system, boosting emotional well-being, alleviating job-related stress, and enhancing academic performance. This study further clarifies forest therapy as an emerging and effective intervention to reduce stress levels among college students, particularly when carried out continuously in easily accessible campus forest environments. Such therapeutic activities could serve as a component of daily stress-relieving programs for college students. This assessment offers valuable information for college students, educational institutions, and policymakers to promote the development of forest therapy on university campuses. However, some of the studies included in this investigation lacked methodological rigor. Future research should employ rigorous study designs to assess the long-term impacts of various forest therapy approaches on the mental and physical health of college students and identify the primary influencing factors. This will aid in determining suitable content, forms, and strategies for forest therapy projects tailored to college students, thereby maximizing the potential benefits of forest therapy on their mental and physical well-being.

**Keywords:** forest therapy; physical and mental health; college students; systematic review





## 1. Introduction

In contemporary urbanized societies, acute and chronic stress, along with insufficient stress recovery, are growing concerns leading to long-term health implications. Stress represents a notable public health challenge linked to mental health conditions like burnout syndrome, as well as cardiovascular, gastrointestinal, immune, and neurological disorders [1]. In this context, mental health has become an increasingly urgent public health issue, highlighted by the rising incidence of mental disorders like depression and anxiety. This mental health crisis is particularly concerning within university campuses. Another public health issue is the lack of physical activity among individuals, and college students, due to insufficient exercise, face a higher risk of metabolic syndrome [2,3].

The university years symbolize a pivotal transition from adolescence to adulthood [4], during which students strive for autonomy from their parents. In today's highly competitive society, students face various pressures stemming from academics, social interactions, and employment, which significantly impact their physiology, psychology, and behavior [5]. Research indicates that students experience severe deterioration in mental health due to academic pressure, relationship issues with peers and partners, economic problems,

academic expectations, and uncertainties about the future [6]. In recent years, there has been a rise in reported symptoms of mental health issues among students. In most countries [7–10], more than half of college students undergo varying levels of stress, anxiety, or depression [11]. Students grappling with mental health issues typically exhibit weaker relationships with peers and faculty, reduced involvement in campus clubs and activities, lower academic performance, decreased graduation rates, heightened alcohol consumption, and elevated risks of drug abuse and suicide compared to their counterparts without such issues [7,12–15]. In a study of Emory University students, 11.1% reported having suicidal thoughts within the last 4 weeks, while 16.5% reported a history of suicidal behaviors or self-harm throughout their lives [16]. In a randomly selected group of 8155 students from 15 universities in the United States, 6.75% admitted to having thoughts of suicide, while 0.5% disclosed attempting suicide within the last year [16]. Stress also contributes to stress-related illnesses among students [17]. The impact of academic stress extends beyond the students themselves to the entire society, as an entire generation is susceptible to mental and physical illnesses; this not only impairs societal development but also drains societal resources.

Moreover, studies indicate that only a small portion of students dealing with stress-related mental health problems actively seek out treatment. Garlow et al. [16] reported that only 15% of students with moderate to severe depression or suicidal thoughts in their sample were undergoing treatment.

The university years are a crucial period for promoting health, preventing diseases, and shaping later lifestyles [18,19]. This period is fraught with considerable stress, underscoring the importance of implementing effective preventive measures to ensure students receive essential support throughout their academic journey. This facilitation enables them to successfully navigate their degree programs and transition seamlessly from university to the workforce. Forest therapy emerges as one such measure, recognized for its efficacy in alleviating stress-related ailments like hypertension and anxiety [20].

The natural environment is being increasingly acknowledged as an efficient method for dealing with stress [21]. Forests, a crucial component of nature, are considered a fundamental health resource and play a role in alleviating stress and inducing physiological relaxation [22,23]. Forest therapy is an activity that utilizes the forest resources in the natural environment for health management and rehabilitation treatment. By engaging in activities such as outdoor recreation, forest bathing, and walking in the woods, individuals can promote both physical and mental well-being, improve their quality of life, and prevent diseases through close contact with nature. When individuals are exposed to forest environments, the forest contributes to the restoration of physical and mental health through "five senses experiences" (visual, auditory, olfactory, tactile, and gustatory) [24]. Venturing into parks or forests has been linked to increased happiness, improved moods, and decreased levels of stress, anger, and depression [25]. Regular participation in forest therapy activities provides a natural and relaxing environment, which helps alleviate stress, anxiety, and depression, thus enhancing human health. This includes lowering blood pressure in hypertensive patients, reducing activity in the sympathetic nervous system, enhancing activity in the parasympathetic nervous system, and activating natural killer (NK) cells [26–28]. As pointed out by Song et al. [20], the purpose of forest therapy is to achieve a state of physiological relaxation through stimuli from the forest environment, thereby generating preventive medical effects, improving fragile immune functions, and preventing diseases. Therefore, forest therapy, an alternative method to alleviate stress and provide a sense of relaxation, may prove to be an effective approach for college students to manage academic pressure.

Despite the growing recognition of the health advantages associated with forest therapy and its widespread adoption among university students, there has yet to be a systematic review of the evidence regarding the effects of forest therapy on the physical and mental health of students. However, such a review is essential for enhancing the utilization of forest therapy programs in the future.

The present study focuses on college students due to the various pressures they face from academic, social, and employment domains in today's highly competitive society, which adversely affect various physiological, psychological, and behavioral aspects. Increasing scientific evidence suggests that forest therapy can improve overall health and is an economically effective stress management method. Forest therapy is an emerging activity that combines human health with the forest environment. In recent years, it has received widespread attention due to its positive effects on both physical and mental health. Its role in college students is also evident. The objectives of this study are as follows: (1) to consolidate the effects and roles of different forest therapy activities and locations and assess the content, formats, and approaches suitable for forest therapy programs for college students; (2) to provide a comprehensive overview and synthesize evidence demonstrating the impact of forest therapy on the physical and mental well-being of college students; and (3) to evaluate research methods in existing research. The objective is to provide guidance for future research on evaluating the effects of forest therapy on college students experiencing stress. In this review, forest therapy is defined as engaging in various therapeutic activities while visiting forests or being in forest environments to enhance an individual's health and well-being.

## 2. Materials and Methods

### 2.1. Literature Search

The Preferred Reporting Items for Systematic Reviews and Meta-Analyses (PRISMA) statement was utilized as a guideline for this study's completion [29]. The literature review was conducted using six databases, namely CINAHL, EMBASE, Medline, PsycINFO, PubMed, and the Web of Science (core collection) to identify relevant studies published up to October 2023. The search strings comprised two elements: forest therapy interventions and college students. Three key terms were used for the forest therapy intervention element: "forest therapy" or "forest bathing" or "shinrin-yoku". Regarding the element related to college students, keywords such as "university students", "college students", "academic students", and "students" were utilized.

### 2.2. Selection Criteria

The initial assessment of eligibility was performed by Mei He, who holds certifications as a forest therapy guide from the Association of Nature and Forest Therapy Guides, as well as a psychological counselor from the Chinese Academy of Sciences. Mei He conducted this assessment by reviewing titles and abstracts. Subsequently, five authors (Yuan Hu, Ye Wen, Xin Wang, Yawei Wei, and Gonghan Sheng) independently screened the full texts of 21 articles based on the following criteria: (i) forest therapy activities involve incorporating exposure and/or physical exercise within a forest environment, alongside other structured activities and cognitive-behavioral therapies. These may include forest walking, forest meditation, and engaging with the forest through sensory activities; (ii) studies conducted in real or virtual settings in any country that (1) evaluated the impact of forest therapy interventions on the physical and mental well-being of university students, (2) incorporated at least one control group or condition, (3) underwent peer review, and (4) were published in the English language. Any discrepancies were resolved through discussions among all authors led by the corresponding author (Guangyu Wang).

Mei He's keyword search yielded 288 individual studies, and the search strategies are detailed in the Supplementary Materials (Supplementary Tables S1–S7). A manual check of the references in the retrieved relevant review studies was conducted to find additional manuscripts related to forest therapy programs and college students (Supplementary Table S8). With the assistance of ENDNOTE X9 software, 62 duplicate articles were removed. Following title and abstract screening, 28 individual studies remained. Subsequently, a thorough examination of full-text articles was conducted, adhering to predefined inclusion and exclusion criteria. This process led to the inclusion of 21 studies in the systematic review (Figure 1). Seven articles were excluded from consideration; two

due to non-English language and five because they lacked forest therapy interventions in their methodologies. The selection procedure was overseen by Yawei Wei and Xin Wang, who performed blinded screenings of the studies. Guangyu Wang resolved any conflicts in selection and rendered the final decisions.

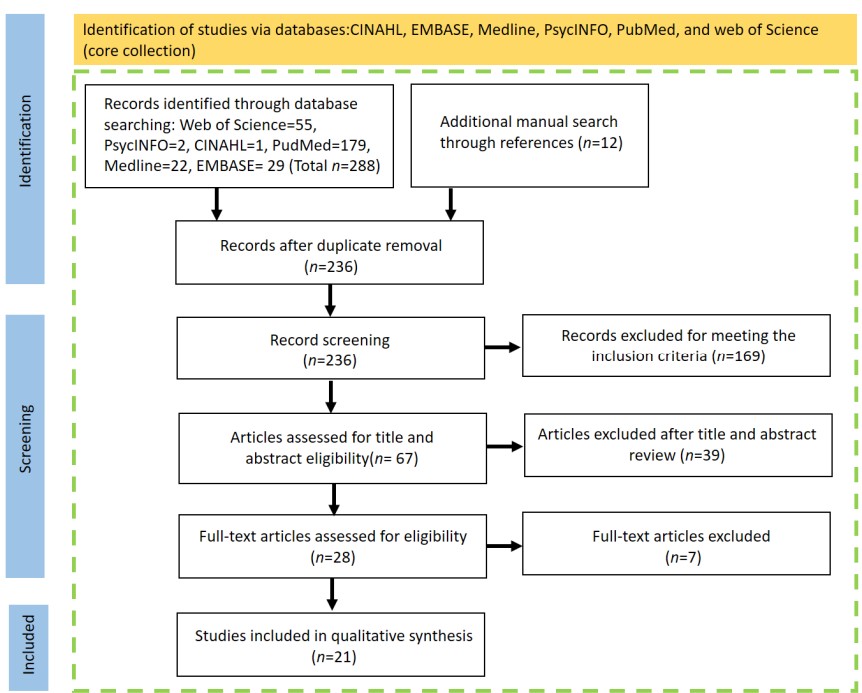

**Figure 1.** The flow diagram of the article selection process.

## 2.3. Data Extraction

Data were extracted using Microsoft Excel v16.0.4266.1001 and then merged into a coding framework. The synthesis was conducted using standardized Microsoft Word v 16.0.4266.1001 (Microsoft Corporation, Seattle, WA, USA) files. The following data were extracted from each study: the author's name, publication year, publication country, study design, participant details, research focus, and main findings. Although a thorough examination and quality assessment of the data were performed, a meta-analysis was not undertaken due to the diverse methodologies employed across these studies.

## 2.4. Quality Assessment Tool

Non-randomized studies underwent evaluation utilizing the Risk of Bias In Non-randomized Studies of Interventions (ROBINS-I) tool [30], which assesses seven domains: confounding, participant selection, intervention classification, deviations from intended interventions, missing data, outcome measurement, and selection of reported results. Randomized control trial (RCT) studies were evaluated using the revised Cochrane Risk of Bias tool for randomized trials (RoB 2) [31]. This tool provides the overall bias risk of randomized trials based on five different domains: the randomization process, deviations from the intended intervention, missing outcome data, outcome measurement, and the selection of the reported results. The internal validity and risk of bias in the studies were assessed. Bias risk was independently assessed by all reviewers and any discrepancies were resolved through a consensus process.

## 3. Results

### 3.1. Study Characteristics

The summary of the studies included in this review is presented in Table 1.

**Table 1.** The characteristics of the included studies (n = 21).

| Author | Country | Number and Demographic of Participants | Research Focus | Key Findings |
|---|---|---|---|---|
| Bang et al. (2017) [32] | South Korea | There were 47 male participants (mean age = 25.5 ± 3.8 years) and 52 female participants (mean age = 23.3 ± 4.3 years) (with 51 participants in the experimental group and 48 participants in the control group). | An initiative introducing forest walks on campus aimed at university and graduate students during their lunch break; the study investigated the program's physical and psychological impacts. | After the intervention, there was a notable rise in health-promoting behaviors ($F = 7.27$, $p = 0.001$, ES = 0.27) and parasympathetic nerve activity ($F = 3.69$, $p = 0.027$, ES = 0.20) observed, alongside a significant decrease in depression ($F = 3.15$, $p = 0.045$, ES = 0.18) within the experimental group compared to the control group. |
| Lee and Koo (2018) [33] | South Korea | 77 out of 82 collected questionnaires were included in the final data analysis. Of these, 49 were from male participants and 28 were from female participants. The sample comprised 29 first-year students (16 male and 13 female), 25 second-year students (18 male and 7 female), and 23 third-year students (15 male and 8 female). | The impact of both pre- and post-plant therapy, which is one of six primary forest healing techniques, on the positive and negative affect, as well as stress levels, among college students of both genders. | (a) The positive and negative affect of college students experienced significant alterations due to plant therapy conducted in a forest setting. (b) In terms of gender disparities, male students saw notable shifts in both positive and negative affect, whereas female students experienced a program that was more effective in reducing negative affect than in enhancing positive affect. |
| An et al. (2019) [34] | China | 13 university students (7 male, aged 21 years). | The effects of physical factor changes in urban forests on blood indices. | The light spectrum and the corresponding changes in temperature and humidity while visiting an urban forest can affect blood pressure. |
| Lyu et al. (2019) [35] | China | 60 male adults aged 19–24 years, with similar health conditions. | The psycho-physiological reactions of participants to the impacts of bamboo forest therapy. | (a) The three-day bamboo forest therapy proved effective in boosting positive mood states and diminishing negative mood states among male participants. (b) After undergoing the three-day bamboo forest therapy session, the male participants experienced a decrease in blood pressure and heart rates (HRs), along with an increase in peripheral oxygen saturation ($SpO_2$). (c) In male participants, bamboo forest therapy led to a notable increase in NK activity and the quantity of NK cells, as well as an increase in cells expressing perforin, granulysin, and granzyme A/B. Additionally, there was a significant decrease in the corticosterone level of peripheral blood lymphocytes (PBLs). (d) The three-day bamboo forest therapy session resulted in improved psychological and physiological well-being and boosted immune functions among the male participants. |
| Rajoo et al. (2019) [36] | Malaysia | 29 students aged 21–23 years (21.83 ± 0.711). | An interdisciplinary investigation aimed at crafting a forest therapy program designed to alleviate academic stress among students. | (a) Following engagement in the forest therapy program, both SBP and DBP values experienced a significant decrease, persisting for up to 5 days before reverting to baseline levels by day 7. (b) Forest therapy presents itself as a valuable resource for students seeking to alleviate academic stress. |

**Table 1.** *Cont.*

| Author | Country | Number and Demographic of Participants | Research Focus | Key Findings |
|---|---|---|---|---|
| Zhou et al. (2019) [37] | China | 43 university students aged 19–23 years. | The geographical difference between urban and rural forest environments, and the effect of urban forest therapy on anxiety alleviation with reference to rural forests. | (a) Engaging in forest bathing can alleviate perceived anxiety, even in intricate details. (b) The impact of forest bathing on the perception of anxiety related to exam pressures and campus life showed geographical variations between urban and rural forest settings.. (c) The greater variety of tree species in the rural forest park led to increased anxiety regarding interpersonal communication. (d) Participants tended to experience greater anxiety relief from social interactions in the urban forest, while experiencing less anxiety related to school work. (e) University students were advised to take a brief visit to an urban forest with companions if they experienced anxiety regarding personal matters and felt the need to engage in conversation with others. |
| Montero and Valverde (2019) [38] | Costa Rica | 50 college students participated in the study, with an average age of $21.88 \pm 4.11$ years. Among them, 27 were physical education students (PEG) with an average age of $21.33 \pm 2.96$ years, and 23 were health promotion students (HPG) with an average age of $22.52 \pm 5.14$ years. | The immediate impact of outdoor exposure on the mood of a cohort of college students in Costa Rica. | Forest therapy indeed proves effective in enhancing the mood of college students. |
| Kim and Lee (2020) [39] | South Korea | 33 fourth-year students at the Department of Nursing of S University in Gangwon-do. | The influence of a forest healing program on job search stress, career decision-making self-efficacy, and career resilience in nursing college students. | (a) The forest healing program yielded a beneficial impact in mitigating negative emotions, specifically employment stress, among the research participants. (b) The forest healing program boosted positive emotions, including career decision-making self-efficacy and career adaptability, among participants. (c) The study confirmed the availability of school forests with easy access to target areas for forest healing programs. |
| Kang and Shin (2020) [40] | South Korea | The experimental group consisted of 35 participants, while the control group comprised 25 individuals. Among these, 29 participants from the experimental group and 11 from the control group participated in a follow-up assessment to evaluate the lasting effects of stress reduction. | The effect of a forest therapy program on academic stress and stress associated with job-seeking among university students. | The experimental group showed statistically significant reductions in both academic stress and job-seeking stress, whereas the control group did not demonstrate similar decreases. |

**Table 1.** *Cont.*

| Author | Country | Number and Demographic of Participants | Research Focus | Key Findings | |
|---|---|---|---|---|---|
| Zeng et al. (2020) [41] | China | 120 university volunteers (60 male and 60 female) aged 19–24 years participated in this study. | The benefits of bamboo forest therapy on the physiological responses of college students. | (a) | Throughout the 3-day bamboo forest therapy session, both observation and walking activities resulted in a decline in blood pressure and heart rate (HR) among university students, accompanied by an increase in peripheral oxygen saturation ($SpO_2$). Among these activities, observation had a more pronounced effect on reducing heart rate in college students. |
| | | | | (b) | During the 3-day bamboo forest therapy session, there was a beneficial impact on reducing systolic blood pressure and heart rate (HR) among college students, particularly notable among female participants. However, levels of peripheral oxygen saturation ($SpO_2$) remained relatively stable. |
| | | | | (c) | The bamboo forest locations efficiently enhanced the physiological well-being of college students, alleviating physical stress and stabilizing their physiological parameters. |
| Bielinis et al. (2020) [42] | Poland, Finland | 42 healthy volunteer students participated in the study, comprising 23 females and 19 males, the mean age of the participants was $26.24 \pm 6.23$ years. | The influence of the forest environment on the subjects' level of procrastination. | Viewing a video featuring forest landscapes could serve as an effective remedy for addressing procrastination issues among students. | |
| Jo et al. (2021) [43] | South Korea | 60 college students (44 male and 16 female). | The impact of virtual reality (VR) forest videos regarding stress levels among college students. | (a) | Forest videos employing VR positively impacted the physiological stress levels of college students. |
| | | | | (b) | Using VR as an alternative for stress management among college students will lead to positive outcomes. |
| Kim et al. (2021) [44] | South Korea | 38 university students ($M_{age} = 22.1 \pm 1.6$ years) were recruited for the field experiment, including 24 male students ($M_{age} = 22.7 \pm 1.4$ years; 63.2%) and 14 female students ($M_{age} = 21.2 \pm 1.4$ years; 36.8%). | The psychological impacts of engaging in forest activities within a campus woodland setting. | (a) | Participants in the intervention group engaging in forest activities reported significant positive improvements in mood, stress response, and subjective well-being. |
| | | | | (b) | Incorporating forest activities within a campus woodland emerges as an effective approach to delivering psychological well-being benefits to college students. |

**Table 1.** *Cont.*

| Author | Country | Number and Demographic of Participants | Research Focus | Key Findings |
|---|---|---|---|---|
| Liu et al. (2021) [45] | China | 30 young adult students. | The rejuvenating impacts of various forest types (mixed, deciduous, and coniferous forests) and nature engagement activities (sitting and walking). | (a) In all types of forests, there were observed benefits in reducing blood pressure and heart rate, as well as decreasing negative emotions while boosting positive ones. (b) The mixed forest demonstrated superior effectiveness in lowering blood pressure and heart rate, while also improving vitality. Additionally, there was a significant enhancement in restoration levels and positive mental health observed in the mixed forest. Conversely, in the coniferous forest, all subscales of the Profile of Mood States (POMS) showed notable decreases, except for vigor. (c) In comparison to the sedentary activity, engaging in walking led to marked decreases in blood pressure and negative emotions, along with significant improvements in restoration, vitality, and positive mental health. |
| Chou and Hung (2021) [46] | Taiwan | 10 college students. | Investigating the impact of regular forest walking on the "cumulative frequency of nature dose". | (a) Engaging in regular forest activities can lead to the accumulation of natural benefits. (b) Through regularly engaging in forest walks lasting at least 30 min once a week for 8 consecutive weeks, participants reported improved mental health, heightened engagement in school learning, increased instances of attention recovery and reflection, and a revitalized connection with nature. (c) Various levels of natural benefits were noted across different cumulative processes. |
| Ibes and Forestell (2022) [47] | U.S. | 234 undergraduate students, including 133 females, who participated in outdoor recreation either frequently or infrequently. | Examining whether students engaged in a passive activity within an urban park-like environment experience enhanced mood, and assessing the potential influence of mindfulness meditation on this effect. | (a) For individuals who regularly engaged in outdoor recreation, spending time outdoors or practicing meditation resulted in a decrease in total mood disturbance (TMD). (b) For college students, especially those who rarely venture outdoors, spending time outside and practicing meditation could serve as effective methods for enhancing mental health. |
| Kotera and Fido (2022) [48] | Japan | 25 students (15 male and 10 female, with a mean age of $20.40 \pm 2.60$ years, ranging from 18 to 28 years) completed assessments at all three points: pre-intervention, post-intervention, and at a 2-week follow-up. | The effectiveness of a 3-day shinrin-yoku intervention study targeting three mental health outcomes: mental well-being, self-compassion, and loneliness. | (a) The average scores for self-compassion, common humanity, and mindfulness showed statistically significant increases from the pre-retreat assessment to the follow-up evaluation. (b) Shinrin-yoku retreats need to be assessed with a broader sample size and within a shorter timeframe to determine the optimal parameters for shinrin-yoku in this context. |

**Table 1.** *Cont.*

| Author | Country | Number and Demographic of Participants | Research Focus | Key Findings |
|---|---|---|---|---|
| Pratiwi et al. (2022) [49] | Indonesia | 32 young university students. | The physiological and psychological effects of walking within a university campus setting. | (a) Walking in a green space on campus led to both physiological and psychological health improvements. (b) Prominent landscape features contributed to the green campus environment. |
| Langer et al. (2023) [50] | Chile | 21 students (with a mean age of 21.4 ± 2.3 years, ranging from 18 to 25 years) from Universidad Austral de Chile (UACh) were randomly divided into two groups: city (n = 11; consisting of 5 female and 6 male students) and forest (n = 10; with 5 female and 5 male students). | Assessed the therapeutic effects of forest bathing on stress and anxiety levels among undergraduate students. | Even a brief session of forest bathing proves effective in alleviating anxiety among undergraduate university students. |
| Shin et al. (2023) [51] | South Korea | 23 university students. | The psychological benefits of a self-guided forest healing program | In comparison to typical activities, both self-guided and guided forest-healing programs markedly improved mood states and alleviated symptoms of anxiety among the participants. |
| He et al. (2023) [52] | China | 36 healthy graduating college students. | The impacts of guided forest therapy on college students. | (a) The guided forest therapy yielded favorable physiological and psychological outcomes for college students. (b) During forest therapy sessions, female participants experienced greater positive physiological benefits compared to male participants, whereas male participants derived greater positive psychological benefits compared to female participants. |

This review includes studies published from 2017 to 2023 (Figure 2), with 95% (n = 19) of them published within the last 5 years. This suggests that the investigation of forest therapy activities for enhancing the physical and mental well-being of university students is a rapidly growing research area. From 2017 to 2018, there was one publication each year, signifying the initial stages of forest therapy application in this context. By 2019, this number increased to five publications each year, demonstrating rapid development. Despite a slight decrease during the COVID-19 pandemic, there has still been a consistent publication rate of between three and four articles annually.

Among the twenty-one studies, seventeen were conducted in Asia, three in the Americas, and one in Europe. South Korea had the highest number of studies in the sample, totaling seven, followed by China with six studies, and then Japan, Poland, Chile, Indonesia, Malaysia, Taiwan, the US, and Costa Rica, with one study each. Regarding the study design, eight studies [33,34,36,38,39,46,48,52] were non-RCTs with pre- and post-tests, while 13 studies employed RCTs.

The studies involved a total of 1161 participants, with participant sample sizes varying from 10 [46] to 234 [47]. Except for one study that focused on nursing college students [39], all other studies involved regular university students. One study had only male participants [35], while the rest of the studies included participants of both genders. Moreover, 90% of the study samples had more than 20 participants. Additionally, about one-third of the studies (38%, n = 8) [33,36,37,39,40,42,46,51] did not mention ethical protocols, e.g., they lacked institutional review board (IRB) review and approval.

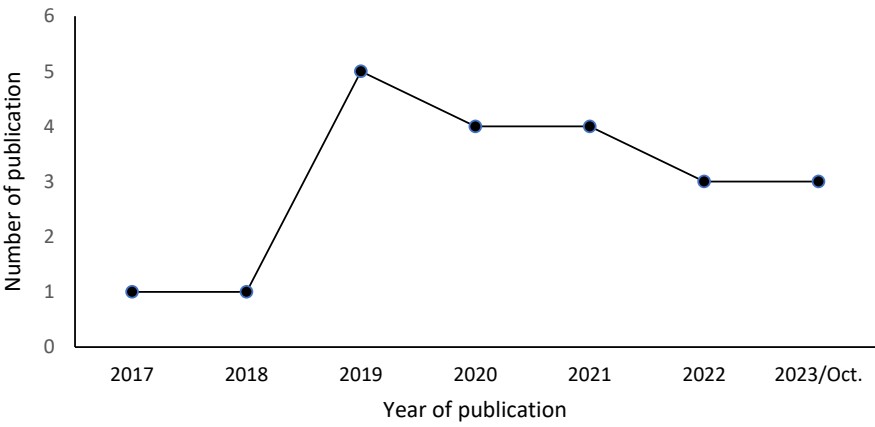

**Figure 2.** The annual number of publications from 2017 to October 2023.

*3.2. Forms and Content of Forest Therapy*

The forest therapy interventions examined in these 21 studies exhibited diversity in both form and content, as outlined in Table 2. The duration of the interventions ranged from 5 min [43] to 8 weeks [40,44,46], with single-day completion being the most common (n = 11). Three studies lasted for 3 days [35,41,48], two studies lasted for 4 weeks [43,51], three studies lasted for 8 weeks, and two studies extended for 6 weeks [32,39]. Among the 21 articles, two focused on virtual reality (VR) forest therapy [42,43], three conducted follow-up evaluations [32,36,48], and three investigated gender differences [33,41,52]. The contents of the forest therapy and control interventions differed across various articles. Forest walking emerged as a primary component of forest therapy and was included in most studies, except for those employing VR therapy through video viewing, sitting, and meditation. The activities included solely walking, walking combined with sitting, walking combined with meditation, and walking combined with other therapeutic activities. Additional therapeutic activities included in the forest therapy programs encompassed immersing oneself in the forest through the five senses (visual, auditory, tactile, olfactory, and gustatory), forest viewing, forest meditation, qigong, aromatherapy, herbal tea therapy, and crafting with natural materials.

**Table 2.** The cross-tabulation of the session duration and content (n = 21).

| Session Content | Composition | Number of Articles | Therapy Environment | Descriptions of Therapy Environment |
|---|---|---|---|---|
| Walking | 40 min once a week for 6 weeks | 4 | Campus forest | The area boasts a variety of tree species, alongside forest roads and trails located near the campus. |
| | 60 min | | Urban forest | The study locations characterized by prevalent tree species. |
| | 15 min | | Campus forest | Academic Event Plaza (AEP) and Landscape Arboretum. |
| | 30 min once a week for 8 weeks | | Campus landscape | Numerous forest trails are present on the campus grounds. |
| Sitting | 35 min | 1 | Natural grass environment | It is enveloped by trees. |
| Meditation | 20 min | 1 | Campus greenspace | A campus resembling a park. |
| Sitting + Walking | Sitting 30 min + Walking 30 min | 1 | National forest park | The forest coverage rate is 86%. |
| Walking + Contemplation | Walking 30 min + Contemplation 15 min | 1 | Arboretum | Rich forest vegetation. |

**Table 2.** *Cont.*

| Session Content | Composition | Number of Articles | Therapy Environment | Descriptions of Therapy Environment |
|---|---|---|---|---|
| Viewing a video | 15 min | 2 | Indoors | A room in the Porthania building. |
| | 5 min per viewing once a week for 4 weeks | | Indoors | The research facility of University C in Cheongju-si. |
| Multiple activities | Briefing 15 min + Forest strolling 15 min + River soaking 30 min + Sensory enjoyment 60 min + Deep breathing exercises 30 min + Group sharing session 30 min | 8 | Forest reserve | Secondary forest reserve. |
| | Freely walking 60 min + Playing a group game 60 min + Having lunch and shopping 60 min + Enjoying the nature views 60 min | | Urban park | The total area is 4.26 km$^2$ and the elevation is between 1100 m and 1400 m. |
| | Getting consent from the forest + Expressing thoughts with ropes + Meditation by breathing in the forest + Walking a woodland path Making a mandala from natural things + Meditation with herbal tea + Qigong functional physical fitness + Dancing therapy + Making a color wheel with natural things + Speaking with value cards + Forest bathing + Meditation with herbal tea for 60 min once a week for 6 weeks | | Campus forest | Surrounded by a variety of herbs and trees, the area incorporates elements of forest healing and is easily accessible to students. |
| | Strolling, forest bathing + Energy massage, dancing therapy + Natural object matching, checking one's emotions with emotion cards + Aroma massage, games using natural objects + Tree climbing + Creating a leaf bouquet + Games with ropes + Making a mandala with leaves for 2 h once a week for 8 weeks | | Campus forest | The site spans an area of 7746.8 m$^2$, with limited access by outsiders, offering expansive views and providing effective screening due to the surrounding trees. |
| | Stretching, respiration, walking, meditation, and exercise for 60 min once a week for 8 weeks | | Campus forest | The campus forest spans approximately 315,000 m$^2$, characterized mainly by diverse vegetation and gentle slopes. |
| | Day 1 involved walking around the area, followed by water activities such as paddle boarding on day 2, and earth activities such as harvesting vegetables on day 3. Each day, participants also dedicated 30 min to meditation and 30 min to yoga. Overall, the intervention spanned 13 h, with 3 h allocated on day 1 and 5 h each on days 2 and 3, respectively | | Scenic spot forest | Rich and diverse nature. |
| | Stretching, diaphragmatic respiration, walking, meditation, and exercise for 60 min once a week for 4 weeks | | Campus forest | Located a brief 10 min walk from the "Isla Teja" campus, it spans over 60 hectares, predominantly consisting of temperate Valdivian forest and areas adjacent to two navigable rivers. |
| | Welcoming 20 min + Pleasures of presence 20 min + What's in motion 20 min + Yoga in forests 30 min + Tea and chatting 30 min | | Regional park | High value as an urban greenbelt. |

**Table 2.** *Cont.*

| Session Content | Composition | Number of Articles | Therapy Environment | Descriptions of Therapy Environment |
|---|---|---|---|---|
| Forest plant healing therapy | 2.5 h | 1 | Campus forest | It stands at an altitude of 400–500 m with a slope ranging from 30 to 35 degrees. Surrounding it is a mixed forest comprising primarily softwood and hardwood, predominantly pine trees. |
| Bamboo forest therapy | 3 days | 2 | Bamboo forest | A sizable variety of clustered bamboo. |
|  | Viewing the landscape for 15 min + Walking in the testing area for 15 min over 3 days |  | Bamboo forest | A large number of species of clustered bamboo. |

Regarding the forest therapy environment, the 21 studies included urban forests, rural forests, protected areas, campus forests, bamboo forests, virtual forest environments, etc. There was a wide selection of forest therapy environments, with the most studied environment being campus forests (n = 9) [32,33,39,40,44,46,47,49,51].

Additionally, of these twenty-one studies, seventeen were conducted with the assistance of forest therapists or professionals trained in relevant knowledge, while the remaining four studies [32,34,41,51] did not specifically mention the involvement of such professionals.

### 3.3. Physiological Impacts of Forest Therapy Programs on College Students

Among the twenty-one studies, nine investigated physiological effects [24,35,36,41,43,45,49, 50,52]. Concerning cardiovascular aspects, the primary factors frequently used to evaluate the impacts of forest therapy included blood pressure, the sympathetic–parasympathetic nervous system, and cardiovascular risk factors. Specifically, eight articles measured blood pressure [24,35,36,41,45,49,50,52], two measured pulse rate [36,52], six measured heart rate (HR) and heart rate variability (HRV) [34,35,41,45,49,52], and three measured peripheral blood oxygen saturation ($SpO_2$) [35,41,50]. A study evaluated immune function by examining NK cell activity, NK cell count, and levels of cortisol, granulysin, perforin, and granzyme A/B in peripheral blood lymphocytes (PBLs) [35]. Another study measured the serum cortisol levels of participants [39], and one study utilized electroencephalography (EEG) to examine changes in the nervous system [10].

### 3.3.1. Cardiovascular Outcomes

The focus of forest therapy interventions in eight studies was stress reduction, with measurements taken for systolic blood pressure (SBP) and diastolic blood pressure (DBP). In two studies [34,50], no changes in the participants' SBP and DBP before and after the intervention were observed. The authors of one study [50] posited that the recruited university student population had good blood pressure levels before the experiment, while the authors of the other study suggested that the lack of change in the participants' blood pressure was due to the short duration of forest bathing (only 15 min), which was insufficient to induce physiological changes. In seven studies, the participants exhibited significant decreases in both SBP and DBP, with one study finding this effect sustained for up to 5 days [36]. Two single-group studies involving pulse rate measurements showed no significant changes before and after the intervention. Out of five studies using HR as a measure of physiological stress, significant reductions were observed post-intervention. For three studies using HRV as a physiological stress indicator, an increase in the high-frequency (HF) component of HRV was observed, indicating enhanced parasympathetic nervous system activity. In studies involving virtual forests, the VR group exhibited more positive effects than the 2D group. Results from three studies on $SpO_2$ demonstrated an increase after the intervention. EEG was employed to analyze alterations in the nervous system, evaluating comprehensive brain-related functions and indicating a decrease in

relative alpha power spectrum (RA) values. Relative beta power spectrum (RB) and the ratio of sensorimotor rhythm from mid beta to theta (RSMT) demonstrated increases in power values, with the VR group exhibiting higher increases in RB and RSMT power values compared to other groups [43].

### 3.3.2. Other Physiological Impacts

Blood markers and NK cells were examined to assess immune function. Findings from a particular study demonstrated that bamboo forest therapy led to a significant increase in NK cell activity, the quantity of NK cells, and the expression of perforin, granulysin, and granzyme A/B ($p < 0.05$). Moreover, corticosterone levels in the participants' PBLs were significantly reduced, suggesting an improvement in immune system function [35]. In another study, blood samples were collected from participants before and after forest therapy to measure fasting serum total cholesterol, low-density lipoprotein cholesterol, high-density lipoprotein cholesterol, and triglycerides. The study revealed no significant changes in these levels before and after forest therapy [32]. One study measured serum cortisol levels before and after forest therapy, and found no significant differences [39]. The researchers from these studies attributed these findings to the already low stress levels among the participants before the therapy.

### 3.4. Psychological Impacts of Forest Therapy Programs on College Students

Among the 21 articles, 86% (n = 18) reported research on mental health, which provides information on the number of studies using specific measurement tools and the reported effectiveness of these tools. Most tools were employed to measure stress (including interpersonal, academic, and job-related stress) and emotions (including positive and negative emotions). The Stress Response Inventory (SRI) was predominantly utilized to assess stress, although some researchers also employed the Maslach Burnout Inventory-Student Survey (MBI-SS), the job-seeking stress survey [40], a clinical data management system (CDMS) [39], and self-written questionnaires. Furthermore, the Profile of Mood States (POMS) (n = 9) [35,38,42,44,45,47,49,51,52] and the State-Trait Anxiety Inventory (STAI) (n = 3) [49–51] were primarily adopted for emotional evaluations. Some researchers also used the inventory of learning self-regulation processes (IPAA) [50], the Self-Compassion Scale (SCS-J), the UCLA Loneliness Scale Version 3 (UCLA-LS3-J) [48], and the Positive and Negative Affect Schedule (PANAS) [33]. The measurement of mental health status encompassed a wide range of indicators, but the tools used were varied. Typically, one or two tools were utilized, such as the Warwick-Edinburgh Mental Well-being Scale (WEMWBS) [45], the Subjective Well-being Mini-Measure (COMOSWB) [44], and the Mental Health Continuum-Short Form (MHC-SF) [46]. One article evaluated depression using the Beck Depression Inventory (BDI). Two studies investigated participants' vitality levels using the Restoration Outcome Scale (ROS) and the Subjective Vitality Scale (SVS) [45]. The research outcomes highlighted stress, emotions, and anxiety as the most significant psychological factors contributing to health promotion.

#### 3.4.1. Emotional and Mood Outcomes

In studies exploring the effects of forest therapy on emotional and affective responses to stress, the Profile of Mood States (POMS) was commonly employed as the primary assessment tool. POMS consists of six domains used to identify and assess transient and fluctuating emotions, namely tension, vigor, depression, fatigue, anger, and confusion [53–57]. Research has found that overall negative emotional states as assessed by POMS significantly improved either partially or entirely following forest interventions, particularly after campus forest activities. Negative emotional states such as tension–anxiety, depression–dejection, fatigue, and confusion experienced notable reductions, while positive emotional states such as vigor exhibited significant increases [35,38,44,45,51].

Regarding anxiety levels, three studies measured emotional and affective outcomes using STAI and revealed significant reductions in anxiety levels among university students

after engaging in forest therapy programs [49–51]. One study, which employed IPAA [50], examined participants' anxiety levels. Meanwhile, other psychological or physiological health indicators did not exhibit statistical differences between pre- and post-intervention (between urban and forest settings) or time effects (within forest or urban treatments), and a potential reason for this could be the short duration of the intervention itself. However, even a relatively short forest bathing session was found to be sufficient to reduce anxiety among undergraduate students.

In a study conducted by [48], Japanese undergraduate students' psychological well-being, self-compassion, and loneliness levels were evaluated using the SCS-J and UCLA-LS3-J after participating in a 3-day shinrin-yoku forest therapy session in Fukushima. Measurements were taken before the intervention, immediately after, and two weeks later. The findings revealed statistically significant increases in average scores for self-compassion, common humanity, and mindfulness from pre- to post-intervention.

Another study [33] used the PANAS to examine the emotional impact on each participant. University students completing forest plant rehabilitation therapy showed a significant increase in positive emotions ($p = 0.000$) and a significant decrease in negative emotions ($p = 0.000$), with male participants exhibiting greater changes.

### 3.4.2. Self-Perceived Stress Outcomes

Five studies included assessments of self-perceived stress, utilizing modified versions of the Stress Reaction Inventory (SRI-MF) [44], MBI-SS [40], CDMS [39], and self- designed questionnaires. These studies encompassed life stress, social relationship stress, academic stress, and job-seeking stress. Studies utilizing modified forms of the SRI-MF indicated significant reductions in overall stress reactions and other SRI-MF subscales, such as "somatic complaints", "anger", and "depression", following campus forest activities. This research confirmed the significant effect of campus forest activities on stress alleviation. One study demonstrated a reduction in academic and job-seeking stress among the forest intervention group [40]. Another study specifically focused on nursing college students [39] and examined job-seeking stress, which was found to be decreased after forest therapy; however, because nursing college students experience lower job-seeking stress compared to general students, the difference before and after the intervention was not significant. These studies collectively suggest that interventions through forest therapy programs can effectively reduce self-perceived stress, particularly significantly impacting the academic and job-seeking stress of university students.

### 3.4.3. Other Psychological Impacts

In a study, the BDI was employed to evaluate students' participation in campus forest walking activities during lunchtime. The findings indicated a notable decrease in depression among the students following the intervention, in comparison to the control group [32]. In a separate study, the Mindfulness-Based Inventory for Students (MBI-SS) was employed to assess academic stress among university students. The findings reported significant reduction in academic stress among participants in the forest intervention group following an 8-week forest therapy program, with this effect lasting for 12 weeks. Conversely, the urban control group did not show significant changes [40]. However, this study did not report intergroup differences in MBI-SS. Two studies [45,48] used the WEMWBS to assess positive psychological health status and revealed increased WEMWBS scores across all forest activities. Two additional studies [42,45] utilized the ROS and SVS to investigate the effects of forest intervention on the vitality of university students. The ROS comprises six items that measure human restoration in forest environments, while the SVS evaluates vitality. Following the forest intervention, participants demonstrated significantly elevated levels of restoration and vitality compared to pre-intervention levels.

### 3.5. Gender Differences in Forest Therapy Programs

Studies examining individual variations in the physiological relaxation effects of forest therapy primarily focus on diverse behavioral patterns, with limited research analyzing distinctions between genders. Among the 21 articles included in the present review, three focused on gender differences in the physiological and psychological effects of forest therapy on male and female university students. One study [41] indicated that forest therapy was beneficial in reducing systolic blood pressure and HR among female participants while maintaining higher $SpO_2$. Another study [33] showed that forest therapy significantly increased positive emotions and decreased negative emotions in men, and the third study highlighted the benefits of forest therapy for both male and female participants. In summary, the results of these three studies all confirm significant psychological benefits for male university students and significant physiological benefits for female university students as a result of forest therapy.

### 3.6. Quality Assessment

The assessment began with an examination of the randomization process to ensure that appropriate methods were utilized, guaranteeing a fair and impartial comparison between the experimental and control groups. Subsequently, attention turned to the evaluation of blinding procedures, investigating whether researchers were aware of the allocation of subjects and whether measures were taken to minimize bias. Another critical aspect scrutinized was the potential for bias during data collection, encompassing the assessment of outcomes and subsequent data analysis.

In the case of non-randomized controlled trials, specific evaluations were conducted to gauge bias risks. Firstly, the presence or absence of randomization was verified, with a subsequent analysis of potential biases if random allocation was not implemented. Additionally, scrutiny was directed towards the selection of the control group, assessing its appropriateness and comparability to the experimental cohort. Finally, measures taken to control confounding factors were examined, ensuring that adequate steps were taken to mitigate any influences that could skew the outcomes of the study.

Regarding the quality of the thirteen RCT studies (Figure 3), eight studies presented a high risk of bias [32,35,37,41,42,45,49,51], one study had an unclear risk of bias [43], and the remaining four studies had a low risk of bias [40,44,47,50]. Regarding the quality of the eight non-RCT studies (Figure 4), six studies showed a high risk of bias [33,36,38,46,48,52], one study had an unclear risk of bias [34], and the final study presented a low risk of bias [39]. The majority of experimental designs offered comprehensive procedural explanations and maintained low participant dropout rates. The sole distinction between the experimental and control groups lay in the treatment method under investigation, thereby bolstering statistical power. Nonetheless, the heightened risk of bias in the randomization process stemmed from inadequate random allocation or the utilization of inappropriate methods, neglecting to adequately address factors that could compromise the validity of the study outcomes. Additionally, a high proportion of bias in the deviation from the established intervention measures was observed due to certain intervention measures affecting the outcomes.

In forest therapy efficacy experiments, the importance of environmental assessment and environmental characteristics cannot be overlooked. Environmental assessment involves comprehensive consideration of the selected forest environment, including factors such as vegetation type, air quality, natural sounds, and climatic conditions. These environmental characteristics are crucial for interpreting study results, as they may directly influence individuals' physiological and psychological responses. For example, whether the forest environment chosen in the study has high-density trees, abundant wildlife, fresh air, and tranquil natural sounds may all have positive effects on participants' emotions, stress levels, and physiological indicators. However, most current studies lack specific records and descriptions of environmental characteristics, and the majority of studies do not analyze the relationship between the environment and therapeutic effects when analyz-

ing results. Therefore, in designing and interpreting forest therapy efficacy experiments, thorough consideration of environmental assessment, environmental characteristics, and their impact is essential.

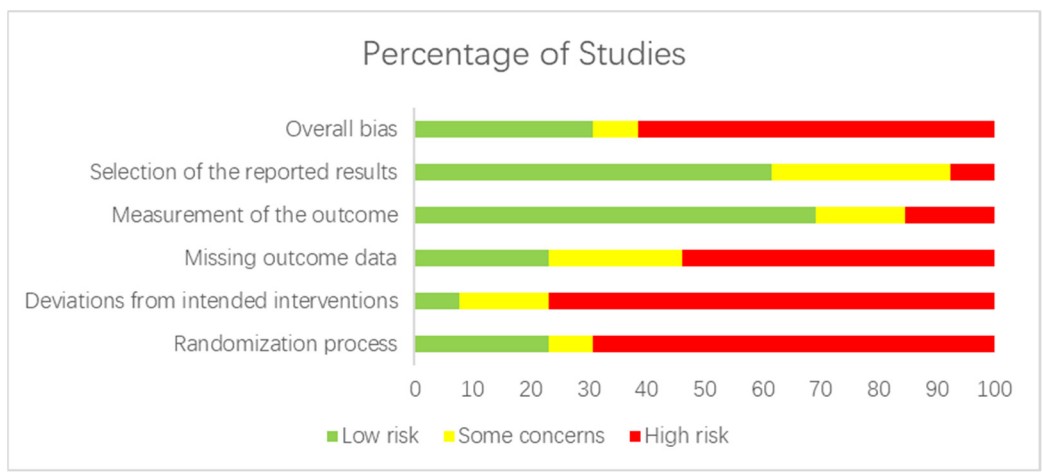

**Figure 3.** The risk of bias graph (RTC studies).

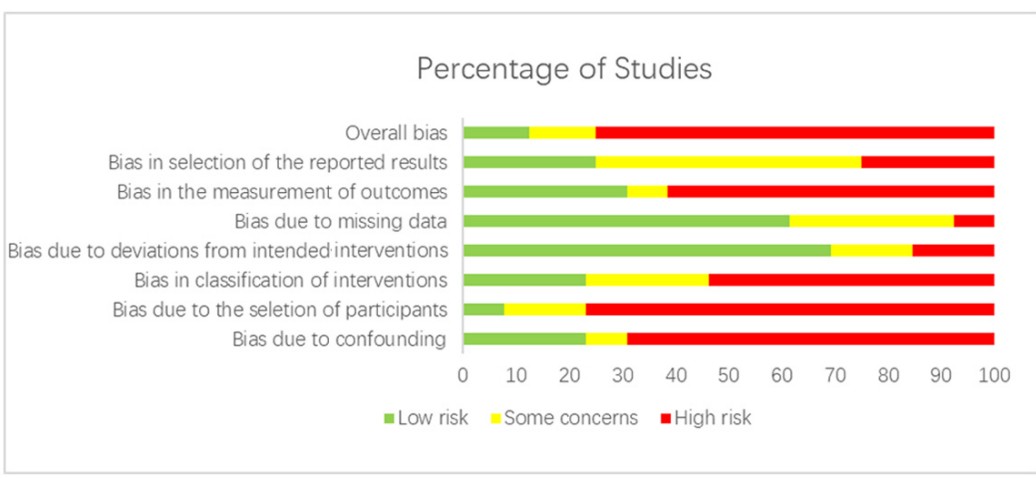

**Figure 4.** The risk of bias graph (non-RTC studies).

## 4. Discussion

Through a comprehensive literature search, 21 studies meeting the criteria for this review were identified, all of which were intervention-based and included at least one control group. Most of these studies were published within the past 5 years, underscoring the emergence of forest therapy as a growing therapeutic approach for addressing the physical and mental well-being of university students, gaining traction in popularity. The interventions across all 21 studies varied significantly in type (such as form, content, and study setting). Despite this diversity, overall, the studies indicated that forest therapy can effectively improve the physiological and psychological health of college students, particularly those experiencing academic and occupational stress. However, while more studies tested the effectiveness of forest therapy, some lacked methodological rigor.

### 4.1. Intervention Measures

Intervention measures for college students based on forest-related wellness activities encompass a wide and diverse range, from individual forest-based interventions, to forest therapy involving various activities within the forest, to complex multi-mode forest therapy programs. Most studies involved forest walking as a form of forest therapy, complemented by stretching, breathing exercises, meditation, physical exercise, and forest crafts, among

others. Walking in a campus forest environment, combined with mindfulness practices and sensory perception exercises, constituted the core elements of forest therapy for college students. Regarding interventions involving multiple combinations of forest wellness activities, it is not currently clear which activities contributed the most to the outcomes or how the different effects of each activity impacted the participants. Further research is needed to clarify and compare the various types of forest activities. Nevertheless, regardless of the form of forest-based wellness activities, they all contributed to the physical and mental well-being of college students to some extent. Most studies suggested that walking and meditation in the campus forest environment are convenient and cost-effective forms of forest therapy that are more favorable to university students. In some studies, forest therapy was combined with elements such as sports activities or therapeutic approaches like psychotherapy, exercise therapy, hydrotherapy, or nutritional therapy, thus forming a comprehensive multi-modal approach to treatment.

Forest therapy programs implemented in diverse forest settings demonstrate positive impacts on the physiological and psychological well-being of college students. Research on and the benefits of campus forests are notably vast and significant. Findings indicate that campus forests serve as an accessible and effective health resource for university students, offering opportunities for relaxation, socialization, and exploration. The present study found that campus forest wellness activities significantly improved the physiological and psychological health of university students. Therefore, the utilization of campus forests for forest wellness activities is the primary choice for forest therapy for students, yielding significant wellness effects that students can easily access at any time. Specifically, campus forests are more suitable than others in creating a healing environment for university students. Additionally, virtual forest wellness activities also have significant therapeutic effects.

*4.2. Health Benefits of Forest Therapy Programs on College Students*

Overall, forest therapy can effectively alleviate the physiological and psychological stress of college students. From a physiological perspective, studies have found that after forest therapy intervention, the systolic and diastolic blood pressure of students significantly decreased under normal conditions, while HR, a physiological stress indicator, significantly decreased, along with improved HRV and increased EEG results. In contrast, forest therapy enhances immune function by boosting the activity of NK cells, decreasing the concentration of the stress hormone cortisol, and regulating the autonomic nervous system. This leads to an improvement in the body's resilience against diseases.

Regarding the psychological effects of forest therapy on college students, findings suggest that it can effectively mitigate psychological stress or mental fatigue, induce positive emotions, and alleviate depression and anxiety, particularly among students experiencing job-related pressures. Depression and anxiety are major challenges faced by the student population, and the psychological effects of forest therapy have proven effective in alleviating various pressures and anxieties in their daily lives. Additionally, forest therapy can effectively improve the sleep quality of college students. Students who undergo forest healing experience a faster recovery of emotions and vitality, resulting in an increase in self-health assessment scores, which is crucial for ensuring their overall health. Another noteworthy outcome is the significant improvement in self-esteem among students after participating in forest healing activities. This is a particularly important psychological trait for the student population, which aids in their smooth transition into society and the realization of their personal value.

Regarding gender differences, forest therapy interventions were found to have different physiological and psychological effects on male and female participants, exerting better psychological therapeutic effects on men and better physiological effects on women. This conclusion can aid male and female students in reasonably choosing forest therapy based on their specific concerns.

### 4.3. Methodological Rigor and Scientific Evidence Quality of the Extant Research

This study included numerous preliminary studies of moderate quality or with a relatively higher risk of bias. The most prevalent issues identified were the absence of blinding among participants, therapists, and/or assessors, which could potentially result in performance and detection biases. Several preliminary studies included in the review also exhibited insufficient reporting quality, such as inadequate descriptions of target groups and randomization, failure to report confounding variables, inadequate reporting of outcomes, and a lack of reporting on dropouts, randomization, and/or blinding. Additionally, these studies suffered from design flaws, including small sample sizes, uneven distributions of characteristics like gender across groups, a lack of representativeness in results, absent or insufficient control groups, a lack of follow-ups, the failure to record essential confounding parameters, disregard for dropouts during the study, the use of unvalidated instruments, and ongoing issues during the research process. Ultimately, dropouts or missing data could pose significant challenges, given that each participant functioned as both the intervention and control group, thereby amplifying the impact of any individual's contribution. However, the carryover effects inherent in the crossover study design were not adequately addressed in the reviewed studies. Only one crossover trial, conducted by the same investigator, mentioned washout periods [51]. In general, the reviewed studies did not address issues related to dropouts or missing data.

### 4.4. Future Research

First, the results of the present review emphasize the necessity for more rigorous RCTs to investigate the effects of forest therapy on the physical and mental health of university students. Future research should aim to improve methodological quality and minimize the risk of bias, thereby enhancing the reliability of evidence-based findings.

Second, the long-term observation of the sustained effects of forest therapy on the physical and mental health of university students is warranted. Post-intervention, the outcomes of multiple follow-up measures (e.g., at 3, 6, and 12 months) should be assessed. This should include various exposure times (e.g., 30, 60, and 90 min) and exposure frequencies (e.g., weekly, every 4 weeks) to pinpoint the optimal duration for the most effective outcomes of forest therapy.

Third, it is essential to examine the optimal effects of different forest environments on the physical and mental health of university students, particularly to leverage the wellness effects and mechanisms of campus forest environments. Furthermore, whether seasonal changes in forest environments significantly impact recovery outcomes requires further research.

Fourth, it is essential to explore which activities and combinations of forest therapy programs effectively enhance the physical and mental well-being of university students. The activities utilized in forest therapy programs are varied, so determining how to customize activities based on the requirements of university students to optimize the health benefits of forest therapy programs requires further investigation.

Fifth, it is imperative for researchers to offer comprehensive explanations of research methodologies and outcomes to yield more dependable results for future interdisciplinary studies, encompassing fields such as environmental and public health.

Sixth, the use of validated tools and research designs suitable for individual studies is necessary for research assessment. The careful consideration of quality assessment tools in the future is essential, as the improper assessment of the quality of individual studies might exaggerate or underestimate the intervention effects.

Finally, given the confirmed positive effects of forest therapy on the physical and mental well-being of university students in this review, the authors stress the importance of extending the implementation of forest therapy among the student population.

*4.5. Limitations of this Study*

This review was characterized by several limitations. The first limitation pertained to the diversity in the measurement techniques, which restricted comparisons between studies and prevented further meta-analysis. Another limitation involved the inclusion of studies conducted only in the English language, thus potentially excluding relevant research published in other languages.

## 5. Conclusions

This systematic review offers an extensive overview of the effects of forest-based preventive and therapeutic interventions on the physical and mental well-being of university students. The review findings indicate that walking in campus forest environments, in combination with mindfulness exercises and sensory perception practices, constitutes a core element of forest therapy for students, serving as a convenient, cost-effective, and noticeably effective therapeutic approach for physical and mental health management. Forest therapy programs demonstrate significant benefits for the physiological and psychological health of university students by (1) protecting the cardiovascular system via reductions in blood pressure and increased NK cell activity and count, and (2) maintaining mental health by alleviating stress and symptoms of depression and anxiety, as well as decreasing negative emotions such as anger.

However, following a rigorous assessment of their methodologies, individual studies were found to exhibit limitations in their study design, control groups, and study populations, which, to some extent, constrains their contribution to the evidence within this research field.

Future researchers should aim to conduct well-designed intervention studies considering the study types, control groups, intervention types and durations, study populations, and descriptions of forest environments. This will enhance the understanding and therapeutic purposes of forest-based health promotion interventions. Further research should focus on the evaluation of the long-term effects of forest therapy, especially in campus forest environments, on the physiological and psychological health of university students. Innovative forest-based wellness activities aimed at mitigating the physical and mental health issues of students, coupled with rigorous research designs, are warranted for empirical studies.

**Supplementary Materials:** The following supporting information can be downloaded at: https://www.mdpi.com/article/10.3390/f15040682/s1.

**Author Contributions:** Writing—original draft preparation, M.H., Y.H. and Y.W. (Ye Wen); writing—review and editing, X.W., Y.W. (Yawei Wei) and G.S.; supervision, G.W.; project administration, M.H. and G.W. All authors have read and agreed to the published version of the manuscript.

**Funding:** This research was funded by Mitacs (Grant No. GR027299), the Basic Research and Talent Research Project of Jiangxi Academy of Forestry (No. 2023521605), and the University of British Columbia (Grant No. GR020223).

**Data Availability Statement:** The data presented in this study are available on request from the corresponding author.

**Conflicts of Interest:** The authors declare no conflicts of interest.

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
