# Peer review of "The Impacts of Forest Therapy on the Physical and Mental Health of College Students: A Review"

_forests, doi:10.3390/f15040682_

Round 1

Reviewer 1 Report

Comments and Suggestions for Authors

Thanks for your paper and comprehensive review. Please find some comments how to improve your in general nice paper.

1. It is unclear what data were presented presented in Figure 1 (Statistics indicating the varying degrees to which students experience stress, anxiety, or depression). Where is depression here? Where stress? Etc. The title of the figure does not correspond with the data presented in the figure. Abbreviation PHQ-4 was not deciphered.

2. Too much information with a lot of details were presented regarding mental health of students on the introduction. Please synthesize the results and present these in a more concise manner. At the same time, specificity of forest therapy was described insufficiently. Please focus on your main object of the study.

3. Please describe clearly how 3.6. Quality Assessment was implemented. I mean the methodology of this procedure/criteria here.

4. One-sentence paragraphs are unwanted.

Author Response

Dear  Reviewer,

Hope you are doing well. We greatly appreciate the time and effort invested by you in evaluating our manuscript titled " The impacts of forest therapy on the physical and mental health of college students: A review" and providing valuable feedback. We have carefully considered all the comments and suggestions provided and have made the necessary revisions accordingly.

Below, we outline the major revisions made in response to your comments:

  1. It is unclear what data were presented presented in Figure 1 (Statistics indicating the varying degrees to which students experience stress, anxiety, or depression). Where is depression here? Where stress? Etc. The title of the figure does not correspond with the data presented in the figure. Abbreviation PHQ-4 was not deciphered.

We want to express that college students are currently facing significant pressure. Based on the feedback from Expert 2, we have removed Figure 1.

  1. Too much information with a lot of details were presented regarding mental health of students on the introduction. Please synthesize the results and present these in a more concise manner. At the same time, specificity of forest therapy was described insufficiently. Please focus on your main object of the study.

As you mentioned, there was indeed an excessive description of mental health issues among college students in the introduction. We have therefore streamlined the description and focused on the physiological and psychological effects of forest therapy on college students.

  1. Please describe clearly how 3.6. Quality Assessment was implemented. I mean the methodology of this procedure/criteria here.

We have included specific evaluative content in 3.6.

  1. One-sentence paragraphs are unwanted.

We have merged or adjusted unnecessary One-sentence paragraphs for coherence.

Thank you for your help again, and have a good day!

Reviewer 2 Report

Comments and Suggestions for Authors

Thank you for the opportunity to review your manuscript. The manuscript is mostly well written but requires clarification/correction in various sections. I have detailed various points in the attached file. 

Author Response

Dear  Reviewer,

Hope you are doing well. We greatly appreciate the time and effort invested by you in evaluating our manuscript titled " The impacts of forest therapy on the physical and mental health of college students: A review" and providing valuable feedback. We have carefully considered all the comments and suggestions provided and have made the necessary revisions accordingly.

Below, we outline the major revisions made in response to your comments:

Abstract

Lines 9-10. “This study thoroughly explores the forest therapy for college students to assess the main findings of previous research.” Revise for clarity of meaning.

We have adjusted the wording to ensure it is logical and coherent.

Article

Figure 1. The figure includes just 4 data points, so provides no additional information beyond what is reported in the figure caption. I recommend deleting the figure. If deciding to retain it, insert axis titles to indicate what X and Y axes represent, and remove decimal places in percentages for Y axis.

Taking into account the feedback from both Expert 1 and your suggestions, we believe that removing the figure as you suggested would be the most ideal approach. 

  • Lines 119-123. With respect to the present study, you note “The goal is to guide future research on the evaluation of the effects of forest therapy on college students facing stress.”However, objective (2) is specific only to the positive impact of FT on college students (“to provide a comprehensive overview and synthesize evidence demonstrating the positive impact of forest therapy on the physical and mental health of college students”), which provides a bias against potential negative effects of FT on college students. How would biased findings help towards future evaluation, which by nature requires balanced perspective taking?

Suer. We should not only emphasize the positive effects, so we have adjusted the expression accordingly.

  • Line 221. You mention that “control interventions varied” but I don’t see these detailed anywhere. Did you do any evaluation of controls? This is a really important factor to determine the veracity of reported improvements’ or ‘enhancements’ because we need to understand these relative to the respective control condition. E.g., a comparison against an urban location as control is not the same as comparing different types of activity (forest bathing vs. something else or guided vs. unguided forest therapy) within the same location. Table 2. With respect to the therapy environment, it would be useful to have some descriptions of the various types of environment. For instance, for those categorised as‘campus forest’, what were the common characteristics? And for each of the outdoor environments, how much did they vary in size and biodiversity? Was there any measure/mitigation of noise and other types of disturbance?

Indeed, different researchers employ varied approaches. Following your advice, we have provided specific details of each forest therapy environment for the readers' reference. Further research is needed to explore the differences in mechanisms.

Line 295. “the maslach burnout inventory-student survey (MBI-SS)”. Does this measure need title case? At least Maslach’s name should be in title case.

Yes, we have already made the modification to it.

Lines 318-319. “Negative emotional states such as “tension-anxiety,” “depression-dejection,” “fatigue,” and “confusion” were found to notably decrease, while positive emotional states like vigor significantly increased [35,38,44,45,51].” Be consistent in using quotation marks for emotional states (e.g., not used for vigor) – they’re not necessary.

Yes, we also think they’re not necessary. We have already made the modifications as per your suggestions.

Line 327. “a potential reason for this could be the short duration of the intervention itself”. Well, in the interests of a balanced perspective for evaluation, we must accept that perhaps a particular forest bathing intervention just didn’t produce the desired effect in some cases or on some measures. According to Table 2, there was a wide range of variability in forest bathing activities and/or location, so we should expect some differences in outcomes arising from that variability.

As we mentioned, there are various approaches to forest therapy, and the specific differences await further investigation in our future research.

Line 379. Again, in the interests of a balanced perspective, I’m a little concerned that you gloss over potential negative effects of forest bathing, and this needs to be addressed. For instance, you summarise benefits only while failing to address that there were apparently also at least some negative effects (i.e., “Another study [33] revealed significant positive and negative emotional effects of forest therapy on men…”). “In summary, the results of these three studies all confirm significant psychological benefits for male university students and significant physiological benefits for female university students as a result of forest therapy.” If the negative emotional effects are such that there was a reduction in negative emotions,that is not a negative effect perse, so I read ‘negative emotional effects’ to refer to an undesirable outcome.

It was an error in our expression. Another study [33] revealed significant positive and negative emotional effects of forest therapy on men…” indicated that forest therapy helped both positive and negative emotions in male university students, promoting positive emotions while reducing negative ones. We have corrected the statement accordingly. Indeed, there is relatively little coverage of the adverse effects of forest therapy at present.

Thank you for your help again, and have a good day!

Reviewer 3 Report

Comments and Suggestions for Authors

Dear authors, thank you for the opportunity to get acquainted with your interesting research.

The strengths of the study are the large coverage of scientific publications on forest therapy and its impact on the psychological well-being and health of students.

The introduction contains detailed information about the relevance of the study for science and practice. It reflects the objectives of the study.

The method is described in detail; a sufficient number of publications have been selected to enable systematic analysis. The search bases used by the authors are impressive.

The results of the study are described in a good structure, illustrated with tables, all explanations are logical and illustrative. After revision, the authors added even more illustrative material and generalizations, which allows us to see the effects of forest therapy on the health and well-being of students in a systematic and comprehensive manner.

The discussion of the results is further developed by the authors discussing in detail the intervention, the health benefits of forest therapy programs for college students, the methodological rigor and quality of scientific evidence of existing studies, and promising directions for future research. Limitations of the present study are presented.

The conclusions are logical and reflect the results obtained.

With best wishes and respect for your work, reviewer

Author Response

Dear Reviewer,

Thank you very much for your support and affirmation of our article. It indeed took us a lot of time and effort to summarize and refine the article on the physical and mental health benefits of forest therapy for college students. We hope to provide relevant information to more people and explore further ways and possibilities for the application of forest therapy in the college student population. Your affirmation and encouragement give us confidence in popularizing forest therapy in the daily lives of college students in the future.

Best wishes,

Mei

Round 2

Reviewer 1 Report

Comments and Suggestions for Authors

Thank you for the improvements. Overall, the iThenticate report revealed a large amount of similarities regarding the results from the other papers, especially in Table 1. It seems that the authors should reword these parts very extensively. These are not standard descriptions and terms, so more should be done in order to prepare this review for publication. 

Author Response

Thank you very much for your assistance and guidance. We have made comprehensive improvements based on the iThenticate report provided by the editorial team and have endeavored to incorporate standard descriptions and terminology to enhance the paper.

Reviewer 2 Report

Comments and Suggestions for Authors

Thank you for providing your revisions and responses to my recommendations. While I am satisfied that you have addressed some of my concerns, I believe there is room for further improvement with respect to critical review and evaluation of the set of studies. I have inserted some further suggestions in the attached file. 

Author Response

My recommendation was intended to indicate the need for additional columns in the table to include these descriptions as well as the comparisons. There is still no reasonable description of a ‘campus environment’, for instance, or a ‘bamboo forest’ in terms of dimensions or other characteristics. If the authors of this set of papers didn’t provide reasonable description of the respective environment, that should be identified in the discussion as a point for critique and evaluation (re your aim 3) and a recommendation for improvement in future research.

There must also be some point of distinction as to what was being compared in each of the studies (e.g., urban location as control is not the same as comparing different types of activity (forest bathing vs. something else or guided vs. unguided forest therapy). These differences must surely have been identified as part of each study’s design.

Table 2 would also benefit from revised formatting of column width according to content. For instance, the Session Content, Number of Articles and Therapy Environment columns could be reduced in width to accommodate a wider column width for the Descriptions of Therapy Environment column.

Thank you very much for your guidance and help. As you mentioned, in forest therapy efficacy experiments, the importance of environmental assessment and environmental characteristics, including factors such as vegetation type, air quality, natural sounds, and climatic conditions, cannot be overlooked.

However, there is indeed a lack of comprehensive assessment of the therapeutic environment in current research. Most articles only provide simple descriptions of the environment, and due to the complexity of the environment, there has been limited investigation into the mechanisms underlying the relationship between the environment and therapeutic effects. This is also a direction for further research that we would like to pursue based on the findings of this review. Therefore, following your suggestion, we have added this issue to the discussion section.

Regarding the format of the table, we have also made adjustments according to the situation.

Your stated aim 3 was “(3) to evaluate the methodological rigor and quality of scientific evidence in existing studies.” Merely accepting that the 21 studies followed different approaches does not speak to any form of evaluation. I don’t consider that this has been adequately addressed.

We primarily analyzed the research methods, so we have modified our wording to align with the focus of our current study. Additionally, following your suggestion, we have included issues and recommendations regarding environmental assessment and analysis, hoping to address this issue in future research as soon as possible.

I further note that the text inserted under section 3.6 Quality Assessment is in bullet point form (without the bullets) without the benefit of any prefatory or explanatory text to indicate that the list represents. Revise the content to narrative format in full sentences.

We revised the content to narrative format with full sentences. Thank you very much for your time again.